# Exploring the burden on family caregivers in providing care for their mentally ill relatives in the Upper East Region of Ghana

Dennis Bomansang Daliri[1,2], Agani Afaya[3,4]*, Timothy Tienbia Laari[5], Nancy Abagye[6], Gifty Apiung Aninanya[7]

1 Presbyterian Psychiatric Hospital, Bolgatanga, Ghana, 2 Department of International and Global Health, School of Public Health, University for Development Studies, Tamale, Ghana, 3 Department of Nursing, School of Nursing and Midwifery, University of Health and Allied Sciences, Ho, Ghana, 4 Mo-Im Kim Nursing Research Institute, College of Nursing, Yonsei University, Seoul, South Korea, 5 Presbyterian Primary Health Care, Bolgatanga, Ghana, 6 Department of Midwifery, School of Nursing and Midwifery, University of Ghana, Legon, Accra, Ghana, 7 Department of Health Services Policy Planning Management and Economics, School of Public Health, University for Development Studies, Tamale, Ghana

* aagani@uhas.edu.gh

**Data Availability Statement:** We have uploaded the data as supplementary information.

## Abstract

Caring for the mentally ill involves numerous challenges, including financial difficulties, stigma, and psychosocial issues, among others. Unpaid family caregivers must endure these challenges as they continue their care for their relatives with mental illness. Despite these burdens and their concomitant effects on both the patients and their caregivers, there is no evidence of this burden in the Bolgatanga municipality. This study explored the burden on family caregivers providing care for mentally ill relatives in the Bolgatanga Municipality of the Upper East Region of Ghana. The study employed a phenomenological research design. Fifteen family caregivers were purposively sampled from two secondary-level health facilities. In-depth interviews were conducted, audio-recorded, and transcribed verbatim. NVivo 12 pro software was used for data analysis. Thematic analysis was conducted following Braun and Clarke's approach. The study identified three themes including social, physical, and psychological burdens. Under social burden, financial challenges and stigma were identified, weight loss was identified as a physical burden, and poor concentration as a psychological burden. These themes represent the challenges encountered by the family caregivers as they provided care for their mentally ill relatives. There is a need to provide support for family caregivers including respite, formation of support groups, and financial support to alleviate family caregivers of the burdens they endure. Additionally, it is imperative to integrate mental health services into the national health insurance scheme to alleviate the financial burden on family caregivers.

## Introduction

Evidence shows that one in every eight people around the world live with a mental illness [1]. As the prevalence of mental disorders continues to rise, so does the demand for care and

**Funding:** The authors received no specific funding for this work.

**Competing interests:** The authors have declared that no competing interests exist.

support for those afflicted. While the healthcare system plays a critical role in providing treatment, a considerable portion of the responsibility for caring for people with mental illnesses falls upon family caregivers [2]. Since the role of a family caregiver in providing support, emotional care, and practical assistance is crucial, it comes with significant challenges and burdens that can have far-reaching consequences on the caregivers themselves [3, 4]. A study in Iran reported a high level of caregiving burden, with 38.2% of the caregivers perceiving severe burden relating to their role [5]. In another study, 27.1% experienced severe burdens while caring for their relatives with mental illness in Iran [6].

Family caregivers are the family members who take care of their relatives with mental illness without payment irrespective of whether they are primary or secondary caregivers [2]. Caregiver burden is the multifaceted strain or negative consequences perceived by the caregiver resulting from caring for a family member with mental illness over time [7, 8]. Caregiver burden can be measured objectively, including the performance of daily assistance activities, financial burden, supervision of patients' behaviour, and disruption of family routine. It can also be subjective, to include feelings of being disturbed during the caring process [9, 10]. Caregivers encounter physical, emotional, social, and financial problems while caring for loved ones suffering from mental illness [7, 11].

Studies have shown that the magnitude of the burden among family caregivers of people with mental illness depends on several factors [12–14]. In Malaysia, family caregivers experienced four forms of negative impact, including financial, social, psychological, and physical [15]. Also, caregivers who were married, less educated, lived in rural areas, had lower monthly income, and provided longer hours of caregiving reported significantly higher caregiver burden in Pakistan [13]. In Brazil, family caregivers who were over 60 years of age, and received no help had higher rates of caregiver burden [14]. In Ethiopia for instance, the age of the caregiver, being a female caregiver, duration of contact hours with the patient per day, perceived stigma by the caregiver, and providing care for patients who had a history of substance use were associated with a higher burden of caregiving [12].

Building on evidence in sub-Saharan Africa (SSA), mental health service resources are inadequate and family members are often expected to provide care for their relatives with mental illness [16]. This responsibility is expected to have been less challenging due to the collectivist nature of African societies [17] since all members in the family are expected to play a role in the care of the mentally ill. This is however not the case since this responsibility is mostly relegated to one person especially women (mothers) [18]. This therefore increases the burden faced by family caregiver responsible for the mentally ill patient in the family. It is therefore not surprising that akin to trends in the Global north which is mainly individualistic [19], family caregivers in SSA also experience extensive burdens including psychological distress [20], and physical, emotional, social, and financial burdens [16].

In the context of Ghana, there are limited resources for mental health service provision. Inadequate infrastructure, limited access to professional care, and stigma associated with mental illness are a few of the challenges encountered by the service [21]. These often result in family members stepping in as primary caregivers. Family caregivers in Ghana provide care across the continuum, from assisting with daily activities to managing medication regimens, accompanying to therapy sessions, and offering emotional support [22].

Albeit, few studies conducted in the southern part of Ghana have reported caregiver burden such as social exclusion, depression, inadequate time for other social responsibilities [22] and stigma from neighbours [23]. While the focus on mental health has grown in recent years, the burden on family caregivers in northern Ghana remains unexplored. This therefore creates a contextual gap in understanding the burden of family caregivers of persons will mental illness in the Bolgatanga municipality. Hence, this study explored the burden on family caregivers

providing care for mentally ill relatives in the Bolgatanga Municipality of the Upper East Region of Ghana. Understanding family caregivers' burden is crucial for developing comprehensive support systems which will be culturally sensitive aimed at alleviating the burden and promoting the overall well-being of both the caregivers and their mentally ill relatives. The research question was: what is the burden on family caregivers providing care for their mentally ill relatives?

## Methods

### Study design

The study employed a qualitative phenomenological research design to explore the burden of family caregivers in the provision of care for mentally ill persons in the Bolgatanga municipality in the Upper East region of Ghana. This methodology was appropriate for the study since it allowed the researchers to investigate the carers' experiences. Additionally, this design explores and emphasizes the "whys" behind certain acts, as well as the "what" and "how" of behaviour [24]. The reporting of the study followed the Standards for Reporting Qualitative Research (SRQR) [25].

### Study setting

The study was conducted in two major secondary-level hospitals in the capital city (Bolgatanga) of the Upper East Region of Ghana. The region has a psychiatric hospital in its capital that is staffed by a psychiatrist and mental health nurses and provides in-patient and out-patient treatment to mentally ill patients. This facility is a new hospital established by the Presbyterian Health Services to provide specialist psychiatric care to the people of the Upper East Region and beyond. It was established and commissioned in October 2022. It acts as a referral mental hospital for northern Ghana, southern Burkina Faso, and parts of the Republic of Togo being the only psychiatric hospital in the region. In addition, the region has a regional hospital in its capital that has a mental health department and staffed by mental health nurses who provide care to mentally ill patients and their relatives. These two hospitals were selected because they recorded the highest number of mental health attendants in the municipality according to records from District Health Information Management Systems (DHIMS-2) and hence had the greatest number of family caregivers accompanying their relatives to these services.

### Study population and inclusion criteria

The study population included family caregivers who accompanied their relatives who were mentally ill patients to the hospital for treatment. Family caregivers who were 18 years or older and agreed to participate in the study met the inclusion criteria. This age category was chosen because this is the age group considered adult and can take up the responsibility of care. This is also the age that can provide consent for the study. However, caregivers who were service providers were not eligible for the study.

### Sampling method and sample size

Purposive sampling was employed to select the participants for the study. Purposive sampling allowed researchers to select participants based on their previous experiences in providing care for mentally ill patients [26]. The first author visited the two health facilities and sought administrative permission before starting the study. Only caregivers who consented to participate in this study were recruited. Data saturation, a condition of data redundancy in which no new

information surfaced, was used to estimate the sample size for the study [27]. On the fifteenth participant, data saturation was attained, and no participant dropped out of the study.

### Data collection tool and method

Unstructured interviews were conducted with family caregivers who consented to participate in the study. This method was employed due to its flexibility in empowering participants to express their experiences and explore matters in detail. We adopted the interview guide used in a previous study by Devkota et al. [28] and modified it to suit the objectives of this study. Follow-up probes were used to ensure that participants shared their experiences regarding the same subject matter. The interview guide included questions such as the participants' experiences with providing care, and difficulties they encounter while providing care. In-depth interviews were conducted from February 2023 to March 2023. The interviews were conducted by the first author who is an experienced health systems researcher with a decade of experience. He is also fluent in both the local language, Grune and English. Interviews were conducted in a secluded nurse's room where the privacy of the caregivers was ensured and devoid of interruptions. Before the interviews, all the caregivers completed written informed permission forms; those who couldn't append their signatures, thumb printed. Each interview lasted around 30–45 minutes, and no repeated interview was undertaken. Fifteen in-depth interviews were conducted in all, out of which four interviews were conducted with participants who spoke Grune as their primary language. In order to allow the caregivers to fully express themselves, open-ended questions with probes were used during the interviews. Field notes of nonverbal cues were obtained while the interviews were audio-recorded. Iterative questioning was used to clarify responses that were ambiguous.

### Data analysis

Thematic analysis was used to analyze the data, adhering to the six-step procedure recommended by Braun and Clark [29]. The first step of the data analysis procedure was the verbatim transcription of the recorded data. The recorded interviews were transcribed verbatim by the first author. For verification of the accuracy of the data, member checking was performed, in which the transcribed data was crosschecked for compatibility with the recorded audiotapes by the first author. The first and last authors were acquainted with the data by reading each transcript numerous times before they were imported into the NVivo 12 pro for analysis. Each transcript was opened in the NVivo software and line by line reading coding was done. These codes were then sorted based on their similarity to the subject matter. The first and last authors reviewed the codes and disagreements were discussed and agreements reached. The next phase in data analysis employed the inductive approach in search of relevant themes based on the data and codes generated. The themes were reviewed to get a better grasp of what was in the transcription. To ensure the themes' correctness, they were verified against the original data to determine their validity and dependability. The themes were then labelled and specified to distinguish one theme from another for accuracy and clarity. The themes were then defined, further refined, and named. A report was then written from the data to present the burden of family caregivers of mentally ill patients in a concise, coherent, and logical manner. The transcripts were anonymized by removing the names of the participants and substituting them with unique codes (FCG1, FCG2, FCG3... FCG15). Soft copies of transcripts were maintained in a flash drive and a password-protected laptop to avoid data loss and simple retrieval.

## Trustworthiness

The methodological rigor was ensured following the criteria outlined by Lincoln and Guba which include credibility, dependability, conformability, and transferability. They later on included authenticity [30]. To ensure credibility, space triangulation, thus, the collection of data from different sites was done through the selection of participants from the Psychiatric hospital and the Upper East regional hospital, both in Bolgatanga, Ghana [31]. Dependability was ensured through peer debriefing and maintenance of the process log to ensure uniformity of the research process [32]. Conformability was ensured by keeping an audit trail of the methodological process. Member checking by participants was also ensured. Transferability was ensured through thick description, thus, a detailed description of the sample characteristics and research setting. Authenticity was ensured through the selection of the appropriate participants for the study with a rich and detailed description [30, 32].

## Ethical considerations

Ethical approval for the study was sought from the Institutional Review Board of Kwame Nkrumah University of Science and Technology with reference number CHRE/AP/038/23. The management of the two hospitals (Presbyterian Psychiatric Hospital and the Upper East Regional Hospital) granted written institutional approval before the commencement of the interviews. All the family caregivers gave written consent and signed the informed consent form before the interviews. All the family caregivers were informed about their right to voluntary participation and withdrawal without consequences. The names of the family caregivers were omitted, and unique codes were assigned to maintain their privacy and anonymity.

## Results

### Demographic characteristics of participants

The study was made up of 15 family caregivers who cared for their mentally ill relatives in the Bolgatanga Municipality. Eight participants were female of which seven were mothers, and seven were male of which four were brothers. Mean age was 43 years (SD 14) (Table 1).

Table 1. Characteristics of participants.

| Family caregivers (FCG) | Age (Years) | Sex | Relationship with patients |
|---|---|---|---|
| FCG 1 | 45 | Female | Mother |
| FCG 2 | 68 | Female | Mother |
| FCG 3 | 40 | Male | Husband |
| FCG 4 | 38 | Female | Mother |
| FCG 5 | 20 | Male | Nephew |
| FCG 6 | 58 | Female | Mother |
| FCG 7 | 26 | Female | Wife |
| FCG 8 | 57 | Male | Husband |
| FCG 9 | 43 | Female | Mother |
| FCG 10 | 59 | Female | Mother |
| FCG 11 | 33 | Male | Brother |
| FCG 12 | 27 | Male | Brother |
| FCG 13 | 56 | Female | Mother |
| FCG 14 | 32 | Male | Brother |
| FCG 15 | 39 | Male | Brother |

## Burdens encountered by the family caregivers

The burdens encountered by the family caregivers of mentally ill patients were classified into three emergent themes and eight subthemes as presented in Table 2 below. Further details can be found in S1 Text.

## Social burden

The participants experienced social burden as a result of providing care for their relatives with mental illness. The social burden associated with the caregiving process was mostly related to caregivers' finances, employment, family, and stigma. This theme is divided into four sub-themes including financial challenges, loss of employment and productivity, loss of family contact, and stigma.

**Financial challenges.**  All participants reported that financial challenges resulting from the high cost of mental health care were a major burden they faced in bringing their relatives for mental health services. Among other costs incurred, the cost of travelling long distances for investigations and care took a heavy toll on their finances.

> *"It has cost me so much with very expensive investigations. I remember I spent 6,000 cedis the last time travelling to Tamale and Accra for investigations" (FCG 1).*

> *"We spent a lot of money travelling up and down for care in Accra and other places" (FCG 3).*

**Loss of employment and productivity.**  The majority of the caregivers were unable to go about their usual jobs as before as a result of taking care of their mentally ill relatives. While the caregiving process cost some of the participants their sources of livelihood, others became unproductive and had to rely on others for financial support.

> *"Another thing that has affected me is that because of his condition, I am unable to get the time to go and work like before" (FCG 4).*

> *"When I was in Kumasi, I was helping with some painting job, and I got my own money but now I have to call people and family members who sometimes do not pick up the phone because they know you are going to ask for money" (FCG 5).*

**Loss of family contact.**  Some participants explained that the care for their mentally ill relatives including bringing them for mental health services has led to their inability to spend quality time with other family members. One participant reported that because of the care for his mentally ill son, he is unable to travel to visit other children outside of the town.

**Table 2. Thematic analysis.**

| Theme | Sub-themes |
| --- | --- |
| Social burden | Financial challenges |
| | Loss of employment and productivity |
| | Loss of family contact |
| | Stigma |
| Physical burden | Loss of appetite and insomnia |
| | Weight loss |
| Psychological burden | Poor concentration/sadness |
| | Excessive worry |

*"I can't leave him alone and go back to my house to take care of my family since he can't take care of himself, with his difficulty to sleep, he calls you at every little opportunity"* (FCG 15).

*"When I want to travel to his other siblings, and leave him alone, he ends up getting missing and because of that I cannot travel"* (FCG 2).

**Stigmatization.** Some of the family caregivers suffered discrimination from the community mainly because they cared for their relatives with mental illness. A family caregiver reported that she has been accused of being the cause of her husband's condition and therefore rain abusive words on her.

*"Also, the people in our community accuse me of being responsible for my husband's illness and say all sorts of bad things about me"* (FCG 7).

## Physical burden

All caregivers experienced some form of physical burden associated with the caregiving. They reported that they have been experiencing medical symptoms as a result of caring for their mentally ill relatives. These medical symptoms were weight loss, poor appetite, and insomnia.

**Loss of appetite and insomnia.** Some of the physical symptoms reported mostly by the family caregivers were loss of appetite and insomnia. They complained that because of the mental illness of their relatives, they at some point lost their appetite and found it difficult to sleep at night.

*"It has affected me severely because, I could not sleep well and could not eat also"* (FCG 13).

A participant reported that because she couldn't eat enough as she should, she had developed peptic ulcer disease.

*"I couldn't sleep, couldn't eat. . .I have now developed stomach ulcer because of the starvation I went through during the time I was following him around"* (FCG 9).

**Weight loss.** Most caregivers experienced weight loss which they attributed to the care of their mentally challenged relatives.

*"I used to be bigger than this but see me now, I have even lost weight. It hasn't been easy for me at all"* (FCG 6).

## Psychological burden

All participants experienced poor concentration, sadness, and excessive worry as psychological burdens they had to endure because of caring for their relatives with mental illness.

**Poor concentration/sadness.** Most of the participants involved in this study reported that they suffered from poor concentration and felt sad most of the time because of their duties as caregivers of their mentally ill relatives.

*"I will be walking but my mind will not be there. Even till now I still struggle. I have body pains, poor sleep and difficulty concentrating"* (FCG 8).

*"Oh, I have suffered a lot, emotionally and financially. . . As for the emotional challenges, they are just too much. I find it difficult to concentrate and mostly feeling sad"* (FCG 1).

**Excessive worry.**   The participants of the study expressed excessive worry as one of the burdens they faced because of caring for their relatives with mental illness. They mostly got worried after seeing the condition of their relatives.

*"Giving birth is a painful experience so I am always worried about my son's condition. It gives me a lot of worry" (FCG 13).*

*"I also think so much causing me sleepless nights because by now she would have been employed and living her own life somewhere if not for this condition" (FCG 6).*

## Discussion

The findings presented in the study highlight the multifaceted burdens faced by family caregivers of individuals with mental illness in the Bolgatanga Municipality of the Upper East Region of Ghana. The identified themes of social burden, physical burden, and psychological burden shed light on the intricate challenges that caregivers navigate in the process of providing care for their mentally ill relatives.

The study showed that family caregivers experience social burdens such as financial challenges, loss of employment and decreased productivity, and loss of family contact. This finding is concordant with Azman et al. [15] who reported that Malaysian family caregivers experienced financial difficulties and disruption of social life as a result of their responsibilities as caregivers. This is further supported by a qualitative study in India [33] and a meta-analysis in which family caregivers of mentally ill patients reported that they experienced financial stress, occupational challenges and family strain [34]. The financial challenges may probably be because of the high cost of caring for the mentally ill persons [35]. Mental health service is not integrated into the National Health Insurance Scheme (NHIS) in Ghana which exposes caregivers and patients to the catastrophic effect of out-of-pocket payments [36]. Also, the loss of employment and productivity identified in this study have been corroborated by Addo et al. whose findings from a systematic review showed that all over Africa, loss of employment and productivity worsened the plight of the family caregiver of mentally ill patients [37]. The roles of family caregivers are likely to result in their inability to fulfil the demands of their various employers leading to loss of employment. In order to reduce the financial burden on family caregivers, the inclusion of mental health services into the NHIS is imperative.

Another burden identified in this study was physical burden. Participants in this study reported poor sleep, weight loss, and poor appetite as the physical burden they had to endure. Findings from Azman et al. [15] reported that family caregivers in Malaysia experienced physical burdens such as poor sleep, poor appetite, and joint pains among others which affirms the current study finding. Walke et al. [38] have suggested that physical burden is the highest burden experienced by family caregivers because of their responsibility of caring for mentally ill patients. The impact of caregiving on family caregivers of mentally ill persons is enormous and causes a lot of physical stress on the caregivers as revealed in this study. The family caregivers' experiences of developing medical conditions, such as peptic ulcer disease, underscore the importance of addressing the holistic health needs of caregivers. Interventions should not only focus on the mental health of the patients but also consider the well-being of the family caregivers.

Finally, in this study, psychological burden on caregivers was evident in the reported symptoms of poor concentration, sadness, and excessive worry. The persistent emotional challenges underscore the need for mental health support for family caregivers. This finding is affirmed by a Nigerian study which identified psychological burden among social, emotional and

cognitive challenges encountered by family caregivers of patients with mental illness [20]. Vadher et al. [39] have reported that the duty of caring for a mentally ill relative is enough predisposition to the development of depression and anxiety. It is further corroborated by a study from Hong Kong which reported anxiety symptoms as psychosocial challenges experienced by family caregivers [40]. In dealing with this challenge, mental health service needs to establish support groups for family caregivers where they can meet and share experiences in dealing with their various psychological challenges. This is likely to reduce their psychological burden of care [41, 42].

### Strengths and limitations of the study

This is the first study that explored the burden faced by family caregivers in the Bolgatanga municipality. The study was rigorous in its analysis and ensured trustworthiness of the data collected. The study also followed the Standards for Reporting Qualitative Research (SRQR) [25] Despite these strengths, the study may have suffered social desirability bias as interviews were face-to-face and caregivers were likely to give "socially appropriate" answers.

### Future research

This current study used a qualitative study design to explore family caregivers' burden in the Bolgatanga Municipality. Despite the findings presented in this study, future research should consider using a mixed method approach in exploring this topic to provide a more comprehensive understanding of the family caregiver's experiences and the associated factors.

## Conclusion

This study highlights the burden of family caregivers of persons with mental illness which include psychological, social, and physical burdens. Financial challenges, stigma and weight loss among others were the major issues identified among family caregivers. There is a need for support groups at various mental health facilities to enable caregivers to share their experiences and benefit from one another's experiences. Also, financial challenges need to be addressed through the inclusion of mental health services in the NHIS to alleviate the financial burden of family caregivers. Furthermore, extending social protection programs, like the Livelihood Empowerment Against Poverty (LEAP) program, which offers cash transfers to vulnerable groups, to family caregivers could be one way to mitigate the costs incurred by those who care for individuals with mental illness. This would help to buffer the effects of the shocks of the direct costs on the family caregivers.

## Supporting information

**S1 Text. Transcribed data.**
(DOCX)

## Acknowledgments

We acknowledge all the caregivers who participated in this study.

## Author Contributions

**Conceptualization:** Dennis Bomansang Daliri, Agani Afaya, Gifty Apiung Aninanya.

**Data curation:** Dennis Bomansang Daliri.

**Formal analysis:** Dennis Bomansang Daliri, Gifty Apiung Aninanya.

**Investigation:** Dennis Bomansang Daliri, Agani Afaya, Nancy Abagye, Gifty Apiung Aninanya.

**Methodology:** Dennis Bomansang Daliri, Agani Afaya, Timothy Tienbia Laari, Nancy Abagye.

**Project administration:** Dennis Bomansang Daliri, Gifty Apiung Aninanya.

**Resources:** Dennis Bomansang Daliri, Agani Afaya, Timothy Tienbia Laari, Nancy Abagye.

**Software:** Dennis Bomansang Daliri, Agani Afaya, Nancy Abagye.

**Supervision:** Gifty Apiung Aninanya.

**Validation:** Dennis Bomansang Daliri, Agani Afaya, Timothy Tienbia Laari, Nancy Abagye, Gifty Apiung Aninanya.

**Visualization:** Dennis Bomansang Daliri, Timothy Tienbia Laari, Gifty Apiung Aninanya.

**Writing – original draft:** Dennis Bomansang Daliri, Agani Afaya, Timothy Tienbia Laari, Nancy Abagye.

**Writing – review & editing:** Dennis Bomansang Daliri, Agani Afaya, Timothy Tienbia Laari, Nancy Abagye, Gifty Apiung Aninanya.

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
