## [Decision Letter · Decision Letter 0]

20 Nov 2023

PGPH-D-23-01718

“I used to be bigger than this but see me now, I have even lost weight”: Exploring the burden on family caregivers in providing care for their mentally ill relatives

Dear Dr. Afaya,

Thank you for submitting your manuscript to PLOS Global Public Health. After careful consideration, we feel that it has merit but does not fully meet PLOS Global Public Health’s publication criteria as it currently stands. Therefore, we invite you to submit a revised version of the manuscript that addresses the points raised during the review process.

We look forward to receiving your revised manuscript.

Kind regards,

Abhijit Nadkarni

Academic Editor

Journal Requirements:

1. We noticed you have some minor occurrence of overlapping text with the following previous publication(s), which needs to be addressed:

- https://doi.org/10.1111/ppc.12151

In your revision ensure you cite all your sources (including your own works), and quote or rephrase any duplicated text outside the methods section. Further consideration is dependent on these concerns being addressed.

2. Please send a completed 'Competing Interests' statement, including any COIs declared by your co-authors. If you have no competing interests to declare, please state "The authors have declared that no competing interests exist". 

3. Please amend your detailed Financial Disclosure statement. This is published with the article. It must therefore be completed in full sentences and contain the exact wording you wish to be published.

4. In the online submission form, you indicated that "Data for the study can be found in the study as excerpts. Full data will be available from the corresponding author upon reasonable request.".

Additional Editor Comments (if provided):

Reviewers' comments:

Reviewer's Responses to Questions

**Comments to the Author**

1. Does this manuscript meet PLOS Global Public Health’s publication criteria? Is the manuscript technically sound, and do the data support the conclusions? The manuscript must describe methodologically and ethically rigorous research with conclusions that are appropriately drawn based on the data presented.

Reviewer #1: Yes

Reviewer #2: No

2. Has the statistical analysis been performed appropriately and rigorously?

Reviewer #1: Yes

Reviewer #2: No

3. Have the authors made all data underlying the findings in their manuscript fully available (please refer to the Data Availability Statement at the start of the manuscript PDF file)?

Reviewer #1: No

Reviewer #2: No

4. Is the manuscript presented in an intelligible fashion and written in standard English?

Reviewer #1: Yes

Reviewer #2: No

5. Review Comments to the Author

Reviewer #1: The authors aimed to describe the burden of being caregivers of patient with mental health disease in Ghana. I think it is a very well-written manuscript, very structured and easy to follow and addressing a clinical topic that I think is of relevance to healthcare professionals in Ghana and elsewhere, even in wealthier countries. The authors describe a mental health service that is clearly under-dimensioned, thus caring for mentally ill patients are for most parts left to family, with severe effect on social, physical and psychological functioning..

I think the study is almost ready for publication. However, a few issues should be solved before publishing.

I think these revisions all can be labelled as minor revisions:

General remarks:

One problem is that pages and lines are not numbered. Thus I will refer to the page that is given in the pdf, not in the originial word document

Title

Title is very good, and representative of the content.

Abstract

Abstract is in accordance with main text

Introduction

Introduction is very well structured and describes the scientific field satisfactory. In this part the authors refer to studies from other countries. Why were these studies selected? Do they have some socioeconomic similarities to Ghana? Or were there other selection strategies that resulted in these references.

In describing the aim of the study, the authors state that similar studies have been conducted in southern Ghana, and not in the northern part. Nevertheless they state that the aim of the project was to study burden of relatives in the whole of Ghana. This should be clarified.

The sentence: P10 To ensure credibility, space triangulation, thus, the collection of data from different sites was done seems incomplete to me. Please clarify the role of «space triangulation» in this sentence.

P10: Participants were mainly females (53.3%) with a mean age of 42.74 ±14.2 and most of them were mothers to mentally ill patients as shown in Table

I would say «mainly» is imprecise, because you recruited almost similar numbers of male and female participants. Suggested rephrasing: Eight participants were female of which seven were mothers, and seven were male of which four were brothers. Mean age was 43 years (SD 14).

Results

Table 2 Thematic analysis.

The authors should clarify whether themes and sub-themes were defined prior to data collection or as part of the analysis. If defined prior to, the authors should not state in Discussion that «Three main themes emerged including social burden, physical burden, and psychological burden».

I’m not quite sure that the label «physical burden» is relevant for the somatic symptoms, because the symptoms, as presented in quotes, are clearly related to emotional/psychological distress.

Phrase: P14 «in tandem» please re-phrase if you mean that the finding is in accordance with, similar to or concordant with

Reviewer #2: Abstract

1. "Care for the mentally ill is saddled with several challenges financial difficulties, stigma, and psychosocial challenges among others."

Consider rephrasing for clarity: "Caring for the mentally ill involves numerous challenges, including financial difficulties, stigma, and psychosocial issues, among others."

2. "It is also imperative to include mental health services in the national health insurance scheme to alleviate the financial burden of family caregivers."

Consider rephrasing for emphasis: "Additionally, it is imperative to integrate mental health services into the national health insurance scheme to alleviate the financial burden on family caregivers."

Introduction

3. The Introduction section of the paper appears to be inadequately written and does not meet the expected standards for journal submissions. The phrasing and structure could be improved for better clarity and coherence, aligning with the rigorous standards expected in academic publications.

Such as

“Evidence shows that one in every eight people around the world lives with a mental illness.” – The authors have provided only one reference for this claim and that too is a WHO key facts. Citing recent studies based on mental illness would be appreciated.

4. The introduction covers a wide range of topics related to caregiver burden, from global prevalence to specific factors influencing burden in different countries. Consider organizing the information in a more structured manner, perhaps starting with a general overview and then gradually focusing on the specific context of Ghana.

5. Use transitional phrases to guide readers through the logical flow of information. For example, phrases like "Building on this evidence" or "In the context of Ghana" can help connect ideas more smoothly.

6. When discussing caregiver burden in Ghana, try to incorporate specific examples or statistics to illustrate the extent of the issue. This can add depth to the introduction and make it more compelling.

7. Clearly articulate the research gap that this study aims to address. While it's mentioned that studies in northern Ghana are limited, explicitly stating the gap and its significance will provide a stronger rationale for your research.

8. The research question is clear, but you may consider providing a brief rationale for why this question is essential and how its answer contributes to the existing knowledge on caregiver burden in Ghana.

Methods

9. Please provide a justification for the age criterion (18 years or older) for family caregivers. Why was this age range chosen, and how does it contribute to the study's objectives?

10. In the sentence, "Fifteen in-depth interviews were conducted in all, including four in the Grune language," it might be more helpful to specify if these four interviews were conducted with participants who spoke Grune as their primary language.

Discussion

11. The integration of previous studies (Azman et al., Addo et al., etc.) is effective in supporting your findings. Consider expanding on the differences or similarities between your study and the cited studies to provide a deeper understanding of the context.

12. Emphasize the policy implications more explicitly. For example, when discussing financial challenges, clearly state the policy recommendations or changes that could alleviate these challenges, such as the inclusion of mental health services in the National Health Insurance Scheme (NHIS).

13. Consider including a brief section on potential avenues for future research. What aspects of caregiver burden could be explored further, and what methodologies might be employed?

14. The authors can end the conclusion with a strong call to action, summarizing the key recommendations and emphasizing their importance for the well-being of family caregivers and mentally ill relatives.

6. PLOS authors have the option to publish the peer review history of their article (what does this mean?). If published, this will include your full peer review and any attached files.

**Do you want your identity to be public for this peer review?** For information about this choice, including consent withdrawal, please see our Privacy Policy.

Reviewer #1: **Yes: **Eivind Aakhus

Reviewer #2: No

---

## [Decision Letter · Decision Letter 1]

19 Feb 2024

PGPH-D-23-01718R1

Exploring the burden on family caregivers in providing care for their mentally ill relatives in the Upper East Region of Ghana

Dear Dr. Afaya,

Thank you for submitting your manuscript to PLOS Global Public Health. After careful consideration, we feel that it has merit but does not fully meet PLOS Global Public Health’s publication criteria as it currently stands. Therefore, we invite you to submit a revised version of the manuscript that addresses the points raised during the review process.

We look forward to receiving your revised manuscript.

Kind regards,

Abhijit Nadkarni

Academic Editor

Journal Requirements:

We noticed you have some minor occurrence of overlapping text with the following previous publication(s), which needs to be addressed:

- https://doi.org/10.1111/ppc.12151

In your revision ensure you cite all your sources (including your own works), and quote or rephrase any duplicated text outside the methods section. Further consideration is dependent on these concerns being addressed.

Additional Editor Comments (if provided):

Reviewers' comments:

Reviewer's Responses to Questions

**Comments to the Author**

1. If the authors have adequately addressed your comments raised in a previous round of review and you feel that this manuscript is now acceptable for publication, you may indicate that here to bypass the “Comments to the Author” section, enter your conflict of interest statement in the “Confidential to Editor” section, and submit your "Accept" recommendation.

Reviewer #1: (No Response)

2. Does this manuscript meet PLOS Global Public Health’s publication criteria? Is the manuscript technically sound, and do the data support the conclusions? The manuscript must describe methodologically and ethically rigorous research with conclusions that are appropriately drawn based on the data presented.

Reviewer #1: Yes

3. Has the statistical analysis been performed appropriately and rigorously?

Reviewer #1: Yes

4. Have the authors made all data underlying the findings in their manuscript fully available (please refer to the Data Availability Statement at the start of the manuscript PDF file)?

Reviewer #1: Yes

5. Is the manuscript presented in an intelligible fashion and written in standard English?

Reviewer #1: Yes

6. Review Comments to the Author

Reviewer #1: This is almost fit for publication. I think new title is more precise, but still, somehow I miss the original.

There are a few phrases, misspelling etc that should be corrected, such as:

p21: "In a separate study..." - reconsider phrasing, use "another" instead?

p21: they face whiles caring - please correct

p21: Researchers have documented -reconsider choice of word, use "showed" instead?

p22: therefore creates a a contextual - please correct

p27: quote: sometimes do not pick because they know you - is translation correct? alternatively: do not pick up the phone

Otherwise I think this is good

7. PLOS authors have the option to publish the peer review history of their article (what does this mean?). If published, this will include your full peer review and any attached files.

**Do you want your identity to be public for this peer review?** For information about this choice, including consent withdrawal, please see our Privacy Policy.

Reviewer #1: **Yes: **Eivind Aakhus

---

## [Decision Letter · Decision Letter 2]

18 Mar 2024

Exploring the burden on family caregivers in providing care for their mentally ill relatives in the Upper East Region of Ghana

PGPH-D-23-01718R2

Dear Dr. Afaya,

We are pleased to inform you that your manuscript 'Exploring the burden on family caregivers in providing care for their mentally ill relatives in the Upper East Region of Ghana' has been provisionally accepted for publication in PLOS Global Public Health.

Best regards,

Abhijit Nadkarni

Academic Editor

Reviewer Comments (if any, and for reference):

Reviewer's Responses to Questions

**Comments to the Author**

1. If the authors have adequately addressed your comments raised in a previous round of review and you feel that this manuscript is now acceptable for publication, you may indicate that here to bypass the “Comments to the Author” section, enter your conflict of interest statement in the “Confidential to Editor” section, and submit your "Accept" recommendation.

Reviewer #1: All comments have been addressed

2. Does this manuscript meet PLOS Global Public Health’s publication criteria? Is the manuscript technically sound, and do the data support the conclusions? The manuscript must describe methodologically and ethically rigorous research with conclusions that are appropriately drawn based on the data presented.

Reviewer #1: (No Response)

3. Has the statistical analysis been performed appropriately and rigorously?

Reviewer #1: (No Response)

4. Have the authors made all data underlying the findings in their manuscript fully available (please refer to the Data Availability Statement at the start of the manuscript PDF file)?

Reviewer #1: (No Response)

5. Is the manuscript presented in an intelligible fashion and written in standard English?

Reviewer #1: (No Response)

6. Review Comments to the Author

Reviewer #1: This is OK for publication now

7. PLOS authors have the option to publish the peer review history of their article (what does this mean?). If published, this will include your full peer review and any attached files.

**Do you want your identity to be public for this peer review?** For information about this choice, including consent withdrawal, please see our Privacy Policy.

Reviewer #1: **Yes: **Eivind Aakhus
